# Formative Development of ClockWork for the Postpartum Period: A Theory-Based Intervention to Harness the Circadian Timing System to Address Cardiometabolic Health-Related Behaviors

**DOI:** 10.3390/ijerph20043669

**Published:** 2023-02-18

**Authors:** Rachel P. Kolko Conlon, Haomin Hu, Andi Saptono, Marquis S. Hawkins, Bambang Parmanto, Michele D. Levine, Daniel J. Buysse

**Affiliations:** 1Department of Psychiatry, School of Medicine, University of Pittsburgh, Pittsburgh, PA 15213, USA; 2Department of Health Information Management, School of Health and Rehabilitation Sciences, University of Pittsburgh, Pittsburgh, PA 15213, USA; 3Department of Epidemiology, School of Public Health, University of Pittsburgh, Pittsburgh, PA 15213, USA

**Keywords:** postpartum period, health behavior, eating, sleep, circadian rhythms, body weight

## Abstract

Individuals with body mass index (BMI) ≥ 25 kg/m^2^ before pregnancy have greater difficulty losing the weight gained during pregnancy, and this postpartum weight retention predicts higher risk for cardiometabolic disease. The postpartum period involves substantial disruptions in circadian rhythms, including rhythms related to eating, physical activity, sleep, and light/dark exposure, each of which are linked to obesity and cardiometabolic disease in non-pregnant adult humans and animals. We posit that a multi-component, circadian timing system-based behavioral intervention that uses digital tools—*ClockWork*—will be feasible and acceptable to postpartum individuals and help promote weight- and cardiometabolic health-related behaviors. We provide data from stakeholder interviews with postpartum individuals (pre-pregnancy BMI ≥ 25; *n* = 7), which were conducted to obtain feedback on and improve the relevance and utility of digital self-monitoring tools for health behaviors and weight during the postpartum period. Participants perceived the ClockWork intervention and digital monitoring app to be helpful for management of postpartum weight-related health behaviors. They provided specific recommendations for increasing the feasibility intervention goals and improving app features for monitoring behaviors. Personalized, easily accessible interventions are needed to promote gestational weight loss after delivery; addressing circadian behaviors is an essential component of such interventions. Future studies will evaluate the efficacy of the ClockWork intervention and associated digital tools for improving cardiometabolic health-related behaviors linked to the circadian timing system during the postpartum period.

## 1. Introduction

Emerging data in humans and animals suggest that key behaviors linked to obesity and health are regulated by the circadian timing system (CTS), or biological clock. The CTS synchronizes internal timing with the natural light-dark cycle and regulates the circadian (approximately 24-h) rhythms observed in most physiological and behavioral measures [1]. Circadian rhythms have well-documented impacts on weight and cardiometabolic health [2], including metabolic processes, energy expenditure, and insulin sensitivity [3]. Alterations in the timing and regularity of circadian rhythms, including sleeping and eating, are related to obesity and weight management [3]. Circadian rhythms can be disrupted by a host of biological, social, and environmental factors. The postpartum period, or the period beginning immediately after childbirth and through the first year postpartum, is a time during which disruptions in each of these realms may be particularly salient.

Health behaviors can rapidly fall out of synchrony with the 24-h light-dark cycle during the postpartum period, as the zeitgebers (time cues) that usually regulate such biopsychosocial rhythms are disrupted [4,5]. Postpartum individuals experience significant challenges with CTS-related behaviors [6], including eating, physical activity, sleep, and light/dark exposure. Of note, we recognize that not all individuals who give birth identify as women, and use the term “birthing individual” to include people who give birth, women, and femmes. Thus, the postpartum period is a critical window for addressing birthing individuals’ weight-related behaviors and health [7], and we posit that the postpartum period offers a unique window for a circadian-based behavioral intervention. We propose that an intervention using the CTS as a unifying framework, with digital tools to facilitate engagement, is a novel approach to addressing the critical issue of postpartum weight retention (PPWR).

Compared to those with normal weight prior to pregnancy (body mass index [BMI] < 25 kg/m^2^), individuals who begin pregnancy with a BMI ≥ 25 kg/m^2^ are more likely to exceed guidelines for gestational weight gain [8,9] and to retain a larger amount of weight postpartum [10,11,12]. Substantial PPWR, defined as a postpartum weight that is at least 5 kg higher than pre-pregnancy, occurs among 20–50% of birthing individuals at six months postpartum and among 25–60% of birthing individuals at 12 months postpartum [13,14,15]. Higher rates of substantial PPWR are observed among people who began pregnancy with a BMI ≥ 25 kg/m^2^ [13,14,15]. These weight patterns reflect retention of the weight gained during pregnancy as well as weight gain that occurs postpartum [13,15,16,17]. PPWR increases risk for cardiometabolic disease directly, as well as through the behavioral factors, such as eating behaviors, that contribute to higher weight retention and/or weight gain in the postpartum period [15,18,19].

Furthermore, the postpartum period is associated with unique biological and psychosocial changes that increase risk for weight retention and present challenges to birthing individuals’ engagement in behavioral interventions. For example, fluctuations in weight may be related to breastfeeding, hormonal changes, reduced sleep, increased stress, and depressive symptoms [15]. These factors not only increase risk for PPWR, they also challenge individuals’ engagement in weight management interventions. For instance, intensive self-monitoring, the hallmark of effective weight management interventions, can be difficult for postpartum individuals and in person sessions can be complicated to attend. Thus, safe, accessible, and effective interventions that address unique considerations of the postpartum period are needed.

Unfortunately, existing postpartum weight interventions have moderate efficacy and low retention rates [20,21,22]. Birthing individuals who are younger, from racially minoritized groups, or from lower income groups are less likely to participate in such postpartum interventions [23], highlighting the need to create accessible, effective interventions to promote health behaviors in ways that postpartum people can engage with and sustain. A meta-analysis of studies among birthing individuals with overweight/obesity indicates that brief interventions targeting eating and physical activity delivered early postpartum shift behaviors, with mixed findings regarding reductions in PPWR [24]. Similarly, brief interventions for sleep may improve the fragmented pattern of postpartum sleep and hasten the return to a more consolidated pattern [24,25,26,27]. Although both eating and sleep behaviors predict PPWR [18], most postpartum weight-related interventions target eating and/or physical activity behaviors alone [15] and rarely target additional behavioral domains such as sleep, which might explain the limited effects of existing postpartum interventions [18].

Notably, CTS-related behaviors that affect cardiometabolic health interact with one another. For instance, when people have shorter sleep duration and poorer sleep quality, they make less healthy food selections, including higher-calorie options, and eat more in studies of sleep restriction [28]. In addition, circadian rhythms related to sleep and metabolism affect and are affected by food intake patterns, especially as the timing of food intake is a zeitgeber, or cue, for circadian rhythm regulation [29,30,31]. However, most CTS-related interventions focus primarily on one behavior domain (e.g., sleep or eating patterns), do not address multiple dimensions of target behaviors (e.g., what, how much, and when people eat), and are conducted among individuals who are not at heightened risk for cardiometabolic disease (e.g., healthy participants recruited who have a BMI between 18.5–24.9 kg/m^2^). Thus, addressing multiple behaviors that are influenced by the CTS has potential to improve weight management and cardiometabolic health, including among individuals at high risk for cardiometabolic disease, although no CTS-based behavioral interventions have been developed for the postpartum period to date.

Thus, it is biologically plausible that focusing on multiple circadian-linked behaviors during the postpartum period may attenuate PPWR. Interventions in non-postpartum populations that address circadian misalignment and eating timing have demonstrated promising effects on weight, glucose metabolism, appetite regulation, and other cardiometabolic health-related factors [3,32,33]. More specifically, time-restricted eating interventions, which address eating windows (in relation to fasting intervals) and the timing of eating, improve body weight and metabolic outcomes [34]. Recent work has suggested that targeting sleep, physical activity, and sedentary behavior across the 24-h day using a digital app intervention is feasible and efficacious among military veterans [35]. However, more data are needed on ways to optimize CTS-based interventions to facilitate sustainable health behavior change and promote cardiometabolic health.

Our unique solution to addressing postpartum cardiometabolic health-related behaviors and PPWR is a CTS-based intervention package: ClockWork. ClockWork is a behavioral intervention that utilizes our knowledge of the CTS to jointly optimize key behaviors related to weight and cardiometabolic health and incorporates digital tools to facilitate health behavior change. In the present study, we assess the intervention’s feasibility and acceptability, and obtain feedback from postpartum individuals on the necessary modifications to improve the accessibility and impact of the intervention. Below, we present results from interviews with postpartum stakeholders as well as the refined ClockWork intervention model.

## 2. Materials and Methods

ClockWork intervention conceptual model. Using the CTS as a unifying framework, the ClockWork intervention is designed to help participants optimize four key behaviors related to weight and health: eating, physical activity, sleep, and light/dark exposure (Figure 1; adapted, in part, from Videnovic et al., 2014 [36]). The four key behaviors are regulated by the biological clock and help to synchronize the biological clock with the environmental light-dark cycle. Multiple dimensions of each behavior are targeted, with an emphasis on the Amount of eating, activity, sleep, and light/dark exposure, and on their Regularity and Timing—the ART of ClockWork.

The ClockWork postpartum intervention has two unique features: (1) targeting multiple behaviors using a common conceptual framework linked to the CTS, and (2) addressing multiple dimensions (i.e., the amount, regularity, and timing) within each behavioral domain. Although many lifestyle interventions for weight management have addressed multiple behaviors, such as eating and physical activity, targeting sleep-related behaviors is less common, and discussion of light/dark exposure or the circadian clock is innovative. For example, if sleep content is included in weight management interventions, they typically only address one dimension of sleep (e.g., duration) or sleep-related environments (e.g., sleep hygiene). Sleep duration has received great attention in cross-sectional and longitudinal obesity studies, although a multidimensional sleep health perspective is emerging as critical for understanding and addressing weight-related outcomes and cardiometabolic disease risk. Similarly, multiple dimensions of health behaviors (e.g., sleep duration, timing, and regularity across days) provide feedback to and are impacted by the CTS. Thus, our inclusion of sleep duration, regularity, and timing to target multidimensional sleep health while using the unifying framework of the CTS is innovative in weight management programming, especially for delivery during the postpartum period.

Sleep-focused interventions may address behaviors related to sleep and light/dark exposure in great detail, with some inclusion of eating and activity-related behaviors to the extent that they affect sleep patterns (e.g., reducing caffeine consumption near bedtime). Time-restricted eating interventions, or other behavioral interventions that address eating timing may also address sleep-related behaviors, although few have been evaluated among individuals with higher weight status. Thus, extensive discussion of four key health behaviors in relation to the CTS and weight management, while focusing on multiple dimensions of each behavior and associations among them, is unique to the ClockWork intervention model.

ClockWork includes 12 sessions of personalized coaching that address these key behaviors and factors relevant to weight management and the postpartum period. The app includes easy-to-use digital tools for self-monitoring to allow for real-time feedback and intervention delivery. In the calendar view, participants are able to see which of the key behaviors they have monitored in a specific time slot to provide a sense of timing across behaviors in relation to the 24-h day. Participants receive 12 weekly tailored coaching sessions to review their digital monitoring of each key behavior, learn intervention content, and set goals. Details on the ART goals for each health behavior are provided below. We also developed a prototype mobile app to monitor the ClockWork target behaviors of sleep, eating, physical activity, and light/dark exposure.

Stakeholder Interview Methods. We used a human-centered design approach to intervention and app development. This approach involves an iterative process that includes obtaining feedback from end-users of the ClockWork intervention; designing and refining the intervention including the digital monitoring tools; and testing participants’ interaction with the intervention model, goals, and app [37]. We used participant interviews to obtain feedback on the ClockWork intervention and digital app prototype. Interviews included open-ended questions and guided walkthroughs, which suggested further design refinements for intervention delivery among postpartum people [37]. Similar methods have been used with other digital interventions designed for the postpartum period (e.g., to address breastfeeding) [38]. We conducted interviews with seven postpartum individuals between May–June 2021. We applied the same sample selection criteria as we envision using in future pilot testing of the ClockWork intervention. Eligibility criteria were: prenatal BMI ≥ 25 kg/m^2^, English speaking, singleton pregnancy. Exclusion criteria were: diabetes prior to pregnancy, using medications known to affect weight (e.g., second generation antipsychotics), current enrollment in weight management programming, recent weight loss surgery (within the past three years), or acute psychiatric symptomatology warranting immediate care. None of the individuals screened for participation in the stakeholder interviews endorsed any of the exclusion criteria. Informed consent was obtained from all subjects involved in the study.

Feedback from stakeholder interviews was used to refine the intervention goals and the app interface, and to improve the engagement with and usability of the intervention content, sessions, and digital monitoring tools. Stakeholder interviews were conducted virtually via videoconferencing with 1–2 participants/interview. Participants were presented with a slideshow that included the rationale for and visual depictions of the ClockWork intervention framework (Figure 1), the ART (i.e., amount, regularity, and timing) of ClockWork health behavior goals, as well as screenshots and a demonstration of the digital monitoring tools for each health behavior in the app (Figure 2). Following the presentation and discussion, participants completed a brief survey to obtain input on the accessibility of their phones or other devices for self-monitoring, the feasibility of the ClockWork intervention goals, and the prioritization of intervention target areas based on interest and perceived effect on weight management. Participants selected one response option for each question, except for the question on barriers to using the app for monitoring, for which they could select all responses that applied. The survey also included demographic questions and the U.S. Household Food Security Survey Module: Six-Item Short Form [39].

We recorded the stakeholder interviews and transcribed the comments. We reviewed the findings in a multidisciplinary team that included: clinical interventionists; experts in weight management, postpartum health behavior change, and multidisciplinary sleep health interventions; and digital health technology experts with app development and design experience. We also characterized the sample using self-reported information from the survey questions and describe the sample characteristics below.

## 3. Results

We completed five interview sessions with a total of seven postpartum individuals before reaching saturation of themes, i.e., repetition of the same feedback and the absence of new themes [40,41]. No further interviews were conducted upon reaching thematic saturation. Sample characteristics from stakeholder interview participants are presented in Table 1.

Qualitative feedback on the ClockWork intervention model for implementation during the postpartum period. The ClockWork intervention was perceived to be helpful for postpartum individuals in managing their weight and related behaviors. Participants reported that the intervention model addresses interconnections among health behaviors and can assist postpartum people with developing a regular 24-h schedule during the early postpartum period while the baby’s sleep is erratic. A screenshot of the slide that was used to present the ART of ClockWork goals to participants for feedback during the stakeholder interviews is shown in Figure 3, and a subset of prompts from the discussion groups and the corresponding feedback provided by participants are shown in Table 2.

*Quantitative feedback on the ClockWork intervention model for implementation during the postpartum period.* Results of the survey responses regarding the ClockWork intervention model and digital monitoring tools are presented in Table 3.

ClockWork intervention model with refinements from stakeholder feedback. Based on the qualitative and quantitative feedback, recommendations for the ClockWork intervention included adjustments of the ART goals to increase feasibility, primarily related to the eating timing goal and sleep timing goal. Participants reported that expanded eating window and midsleep timing goals would be more feasible and that they would be more likely to implement these behavior changes. Table 4 provides an overview of the ART of ClockWork goals and reflects the feedback we incorporated from the stakeholder interviews and survey results. The key changes made to the goals were the shift in the eating timing goal from a 10-h eating window (14-h fasting interval) to a 12-h eating window (12-h fasting interval) and the expansion of the midsleep timing goal from 3:00–3:30 a.m. to 2:00–4:00 a.m.

We organized the order of session content information—eating, multidimensional sleep health, light/dark exposure, and physical activity—to fit with the needs and considerations of postpartum people (e.g., physical activity may not be approved until several months after delivery). In line with participants’ feedback during the stakeholder interviews and surveys, we revised the ClockWork curriculum to introduce the topic of regularized eating first, as it is strongly linked to weight patterns and is a zeitgeber that provides critical circadian scaffolding and cues the implementation of daily routines, including the other key behaviors.

Qualitative feedback on the ClockWork app and digital tools to facilitate self-monitoring in the postpartum period. Participants reported perceived advantages of the ClockWork intervention and digital monitoring tools, including that it would be portable, be easy to use, and reduce the impact of the pandemic on individuals’ health behaviors. Recommendations for the ClockWork app included the following modifications: adding the ability to report baby care activities, increasing the visual contrast and ease of reaching buttons one-handed while caring for the baby, especially during the night, and using engaging prompts, including both gamification as well as reminders (e.g., pop-up windows to reinforce monitoring). The results informed our optimization of the physical activity reporting buttons (Figure 2) and the confirmation buttons (e.g., Report, Save & Cancel buttons) across all of the behaviors. Regarding the burdens of monitoring, one noticeable challenge reported among postpartum individuals was to tap the screen using one hand while holding their baby in the other, regardless of the size of one’s phone. Therefore, we determined that it was important to improve the compatibility of the buttons on the screen, including their distinct coloring, placement, and consistency across behaviors to increase the ease of clicking to complete each monitoring report.

Quantitative feedback on the ClockWork app and digital tools to facilitate self-monitoring in the postpartum period. As shown in Table 3, the majority of participants reported having their phone available and accessible for self-monitoring their behaviors. Results from the survey informed our decision to add prompts in the app to encourage participants to monitor their sleep and light/dark exposure in the morning, rather than in-the-moment at night. There is a high likelihood of performing childcare activities at night in the early postpartum period. Thus, having participants report on their sleep and darkness exposure after final awakening is designed to reduce their screen time during their sleep period and reduce the burden of monitoring while promoting accuracy (e.g., if you report in the app that you are going to bed although your baby begins to cry and you then go to soothe or feed the baby, resulting in a later bed time).

## 4. Discussion

Results of the stakeholder interviews suggest that the ClockWork behavioral intervention and digital tools, designed in collaboration with postpartum individuals, can be accessible anywhere, anytime to support participants and health behavior changes. The primary feedback from the stakeholder interviews indicates that a behavioral intervention package that includes self-reporting via digital monitoring tools has potential to promote postpartum health behaviors and weight management. Additional feedback from the stakeholder interviews related to the importance of eating behaviors, relative to other CTS behaviors, and feedback on realistic goal setting for postpartum individuals was incorporated into the intervention and digital self-monitoring tools.

Participants indicated that the digital tools would facilitate self-monitoring and the tailoring of coaching during intervention delivery. Specific to the postpartum period, we were interested in learning about phone availability given that individuals often are using their hands for baby care activities, and the accessibility of phones for monitoring behaviors in-the-moment or in near-real time. The qualitative and quantitative feedback from participants indicated that their phones are often, if not always, available and that monitoring using a phone-based app would be feasible. Importantly, given that participants also indicated that self-monitoring behaviors is burdensome and time-consuming, our interpretation was that phones are an appropriate device for monitoring and an app is a feasible monitoring tool. The methods for reporting health behaviors need to be easy, reachable with one finger when holding the phone in one hand (e.g., if holding a baby in the other arm), and highly visible and distinct for postpartum people, especially during the night and when exhausted. Thus, we used a simplified monitoring platform and refined the visual display and placement of the monitoring buttons to make the actions related to monitoring as easy, accessible, and accurate as possible.

Results of the surveys demonstrated that postpartum individuals are interested in working on their health behaviors, particularly related to eating and sleep, and view these as important for weight management. We have included the topic of eating behaviors as the first target area in the ClockWork intervention for several reasons. Eating was reported by the majority of postpartum participants as the behavior they viewed as most related to their weight, and this aligns with the robust number of studies supporting links between eating behavior and weight outcomes. Moreover, from a CTS perspective, eating behavior is a zeitgeber that cues daily circadian rhythms, thus providing a meaningful and achievable target for people in the early postpartum period when daily routines are disrupted.

Strengths of the study include the incorporation of stakeholder input from a population that has received little attention in weight management interventions and digital monitoring tools, and use of human centered design principles to inform the methods and data interpretation. Another strength is that we focused on postpartum people who began pregnancy with a BMI ≥ 25 kg/m^2^, a group at high risk for postpartum negative health sequelae [15,18,19]. There also are several limitations to consider. The sample size for the stakeholder interviews was relatively small. However, we obtained similar feedback across the five interviews that were conducted, suggesting saturation of themes, which is typically an indication of obtaining a strong understanding of feedback on a topic from a given population [42]. While thematic saturation was reached, it may be advantageous to gain further input from postpartum individuals after participating in the ClockWork intervention and utilizing the app. Moreover, it would be beneficial to obtain feedback from postpartum individuals from minoritized groups, as there are numerous structural and systemic factors related to postpartum cardiometabolic health [43].

Of note, we had planned to conduct stakeholder interviews in person, although switched to conducting the stakeholder interviews by videoconference during the early COVID-19 pandemic period when there were many shutdowns and disruptions to everyday life. Feedback from postpartum participants included that the ClockWork intervention was both novel and feasible during the COVID-19 pandemic. In addition, participants noted the relevance of the model and digital monitoring tools to post-pandemic intervention delivery as they perceived multiple benefits for postpartum individuals. E-health interventions have shown promise among postpartum people, including minoritized populations, provided that considerations for accessing the technology and support are addressed and implemented effectively [15].

The current app only provides monitoring features and does not include self-review options (self-management) beyond the calendar view in which participants can see that a behavior has been logged in a particular time slot. Participant progress could only be visualized during the intervention sessions through the clinician’s web portal. Further improvements may include real-time self-review for individuals to view summaries of their monitored behaviors and visualization of goal achievement, which may increase self-management and facilitate greater behavior change and intervention engagement [44]. Future work is warranted to identify the effects of including such self-review features within the app as well as in conjunction with other wearable devices that may utilize objective or passive sensing methods and to evaluate which monitoring and goal achievement visualizations are most effective for postpartum individuals.

The most important future research direction is evaluating the efficacy of ClockWork among postpartum individuals. We are currently conducting a pilot randomized trial to assess the initial efficacy of the ClockWork intervention by randomizing birthing individuals with prepregnancy BMI ≥ 25 to receive ClockWork or usual care during the first six months postpartum. Obtaining input on the specific intervention features, delivery, and digital monitoring tools from larger postpartum samples, with objective measures of the key behaviors, and among other populations, would further refine and optimize the ClockWork intervention. Subsequently, future studies are warranted to evaluate implementation of the ClockWork intervention through health coaching systems for efficient, low-cost, scalable delivery during the postpartum period.

## 5. Conclusions

ClockWork is a behavioral intervention comprising coaching sessions as well as digital monitoring tools. ClockWork is novel in that it: (a) simultaneously addresses four key behaviors related to weight, cardiometabolic health, and the CTS, and (b) employs simple principles to address multiple dimensions for each behavior: the amount, regularity, and timing—the ART of ClockWork. Moreover, the ClockWork intervention uses personalized coaching and digital tools to improve treatment tailoring and accessibility for postpartum people. We encourage future research to evaluate the CTS-based intervention and targeting of the amount, regularity, and timing of multiple key health behaviors to support birthing individuals after delivery and promote cardiometabolic health during the postpartum period. Future extensions also are warranted to other periods of life transition when there are significant shifts in CTS-related health behaviors to optimize cardiometabolic health across the lifespan.

## Figures and Tables

**Figure 1 ijerph-20-03669-f001:**
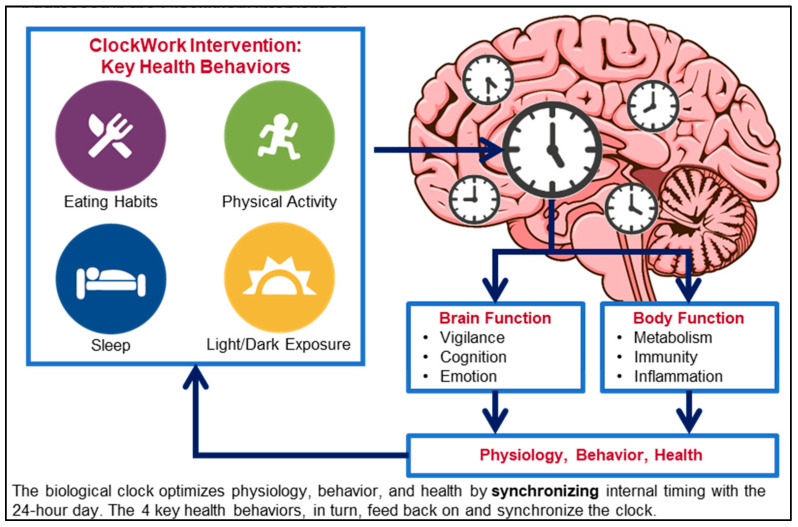
Conceptual model of the circadian timing system and key health behaviors addressed in the ClockWork intervention.

**Figure 2 ijerph-20-03669-f002:**
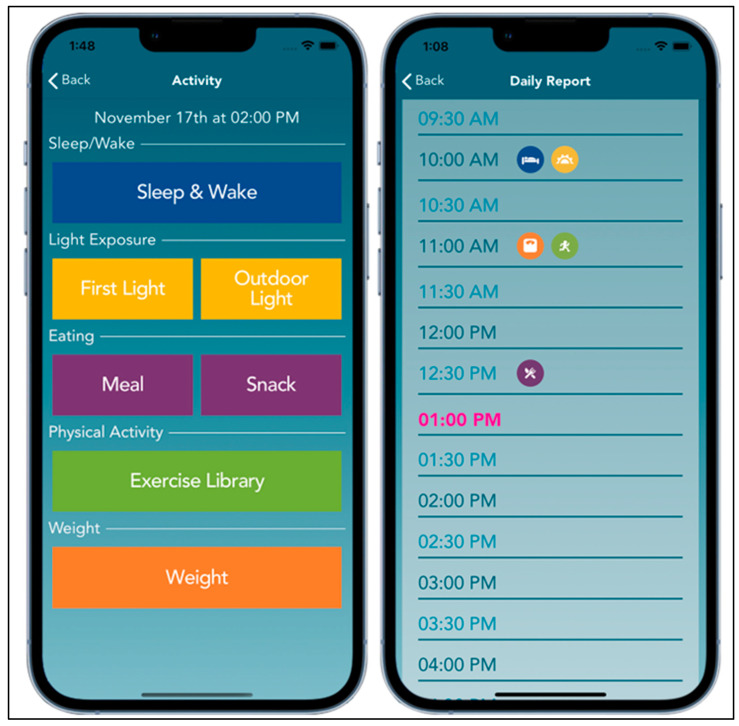
ClockWork digital monitoring prototype.

**Figure 3 ijerph-20-03669-f003:**
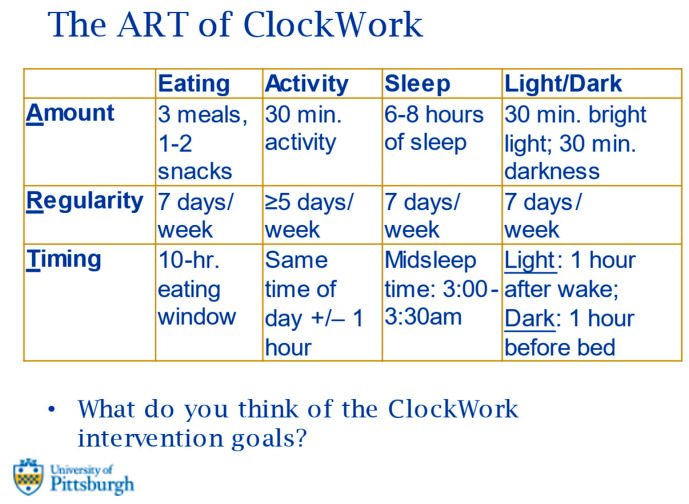
Screenshot of the ClockWork amount, regularity, and timing (ART) goals presented to participants for discussion in stakeholder interviews.

**Table 1 ijerph-20-03669-t001:** Stakeholder interview participant characteristics (*n* = 7).

Characteristic	*n* (% out of 7 Individuals)
Racial/ethnic identity (more than one could be selected)	
Hispanic/Latinx	2 (29%)
White	6 (86%)
Intentionality of most recent pregnancy	
No	0 (0%)
Yes	Yes (100%)
Household income	
$20,001–$30,000	2 (29%)
≥$30,001	5 (71%)
Insurance status	
Employer-provided health insurance	5 (71%)
Medicare/Medicaid health insurance	2 (29%)
Food security	
High or marginal food security	6 (86%)
Low food security	1 (14%)
Relationship status	
Married	6 (86%)
Not in a relationship	1 (14%)
	***M* ± *SD***
Number of pregnancies	2.9 ± 2.0 pregnancies (range: 1–7)
Number of births	2.4 ± 1.5 births (range: 1–6)
Self-reported gestational weight gain	27.6 ± 12.5 lbs. (range: 15–53 lbs.)
Years in current relationship	4.9 ± 1.9 years married (range: 3–8 years)

**Table 2 ijerph-20-03669-t002:** Examples of qualitative feedback on the ClockWork intervention model and digital monitoring tools from postpartum individuals who began pregnancy with a BMI ≥ 25 (*n* = 5 stakeholder interviews with *n* = 7 participants).

Prompt	Examples of Responses Provided
What do you think of the ClockWork intervention model?	“The idea sounds very revolutionary. I like the inclusion of dark/light exposure. Anxiety levels, eating behavior is different between morning and night. Sleep is very interrupted postpartum and is connected to physical activity. If I sleep less, I am less likely to exercise, which leads to more anxiety about eating”. (Stakeholder interview #1)
What do you think of the ClockWork intervention goals?	General comments on the goals across behaviors: “The goals look good. Most people would be able to meet these goals”. (Stakeholder interview #5)
	Eating goals: “These goals are doable. I would do 4 meals and 1–3 snacks. Instead of a 10-h eating window, I’d be willing to do a 12-h eating window”. (Stakeholder interview #1) “A 12 h window is better. Aside from that it’s reasonable”. (Stakeholder interview #3)
	Physical activity goals: “Timing may be difficult due to the baby’s schedule. Sounds reasonable. Activity will need to start 2–3 weeks postpartum”. (Stakeholder interview #3) “It would depend on when after postpartum this starts. Once cleared [for physical activity], I think it would be great”. (Stakeholder interview #4)
	Sleep goals: “Everything would be feasible for me. It may be difficult to put how many times I wake up during the night for the baby. It would be good to have an estimate because I may not remember”. (Stakeholder interview #1) “2–4 a.m. would be best. Seems feasible”. (Stakeholder interview #2)
	Light/dark exposure: “The light goal sounds good”. (Stakeholder interview #2) “Right now, I’m not good with these goals. Me and my husband watch TV before bed because that’s our time together and we sleep with the TV on. I think it’s doable. My room is rarely 100% dark. I play on my phone while I feed [the baby] to stay awake. I cannot stay awake during feedings if there’s nothing to focus on”. (Stakeholder interview #4)

**Table 3 ijerph-20-03669-t003:** Quantitative feedback on the ClockWork intervention model and digital monitoring tools from postpartum individuals who began pregnancy with a BMI ≥ 25 (*n* = 7).

	Item	Responses *n* (%)
ClockWork Intervention	The key behavior that seems most important for you to manage your weight and health postpartum	5 (71%) Eating
		1 (14%) Physical activity
		1 (14%) Sleep
		0 (0%) Light/dark exposure
	Willing to try eating your meals and snacks within a 10-h window (for example, between 8 a.m.–6 p.m.)	4 (57%) No
		3 (43%) Yes
	Willing to try eating your meals and snacks within a 12-h window (for example, between 8 a.m.–8 p.m.)	7 (100%) Yes
		0 (0%) No
	Willing to try exercising for at least 150 min/week	7 (100%) Yes
		0 (0%) No
	Willing to try sleeping for 6–8 h per night	7 (100%) Yes
		0 (0%) No
	Willing to try sleeping so that the midpoint of your sleep is around 3–3:30 a.m. (for example, going to bed at 11 p.m. and waking up at 7 a.m.)	7 (100%) Yes
		0 (0%) No
	Willing to try getting 30 min of bright light within an hour of waking up	7 (100%) Yes
		0 (0%) No
	Willing to try getting 30 min of darkness within an hour of going to bed	6 (86%) Yes
		1 (14%) No
Digital Monitoring Tools	Phone availability	5 (71%) Yes, I have my phone with me at all times
		2 (29%) I have my phone with me most of the time
	Access to phone for monitoring	4 (57%) My phone is always accessible for monitoring
		3 (43%) My phone is accessible sometimes and I could monitor nearly everything
	Preference for monitoring nighttime events (e.g., feeding) in the moment or in the morning	5 (71%) In the morning (remembering or recalling your awakenings, eating, caring for the baby, etc.)
		2 (29%) In the moment
	Anticipated barriers that would get in the way of using the app? *(individuals could select all that applied)*	4 (57%) Monitoring is burdensome
		4 (57%) Too busy/hard to find the time
		2 (29%) Lack of childcare
		1 (14%) Low priority/lack of motivation to manage weight

**Table 4 ijerph-20-03669-t004:** The ART of ClockWork.

	Eating	Activity	Sleep	Light/Dark Exposure
Amount	3 meals 1–2 snacks	30 min. activity	6–8 h of sleep	30 min. bright light 30 min. darkness
Regularity	7 days/week	≥5 days/week	7 days/week	7 days/week
Timing	12-h. eating window	Same time of day +/− 1 h.	Midsleep time: 2:00–4:00 a.m.	Light: 1 h. after wake Dark: 1 h. before bed
Intervention strategies: (1) Self-monitoring behaviors via mobile app; (2) personalized feedback; (3) tailored coaching				

## Data Availability

The data supporting the findings presented in this study are available from the corresponding author upon reasonable request.

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
