# Peer review of "Formative Development of ClockWork for the Postpartum Period: A Theory-Based Intervention to Harness the Circadian Timing System to Address Cardiometabolic Health-Related Behaviors"

_ijerph, 2023, doi:10.3390/ijerph20043669_

Round 1

Reviewer 1 Report

Conlon et al., present a clockwork intervention model that accounts for multiple dimensions of several behavioral domains. 

The unique features of this intervention are the amount, regularity, and timing of several activities such as eating, activity, sleep, and light/dark cycles. This article is of great importance to promoting cardiometabolic health during the post-partum period and even more can be optimized to multiple models and across multiple age groups.

1.     What is the main question addressed by the research? 

Conlon et al., aim to optimize a digital self-monitoring app for post-partum individuals to regulate their daily activities. 

2.     Do you consider the topic original or relevant in the field? Does it
address a specific gap in the field? 

Despite post-partum individuals having disruptions in several behavioral activities, many interventions developed aim at solitary physical activity and thus fall short in assessing overall weight-related outcomes and cardiometabolic risk. Hence it is important to address multiple aspects of physical activities that might provide feedback on healthy behavior. This study aimed at developing and optimizing self-monitoring apps to assess several activities of a post-partum individual. 

3.     What does it add to the subject area compared with other published
material? 

The inclusion of a multidimensional approach and digital tools for self-management is unique and has the potential to promote healthy behavior to post-partum individuals. 

4.     What specific improvements should the authors consider regarding the
methodology? What further controls should be considered? 

Increasing the number of participants could increase the significance of these findings. 

5.     Are the conclusions consistent with the evidence and arguments presented
and do they address the main question posed? 

Yes 

6.     Are the references appropriate? 
Yes 

7.     Please include any additional comments on the tables and figures. 

Figure 3. image quality is poor. 

Reviewer 2 Report

The purpose of this manuscript is to describe a digital tool for improving cardiometabolic outcomes in postpartum individuals with pre-pregnancy overweight or obesity and to report surveys of acceptability and feasibility for tool improvement. The authors present an exciting digital tool for an interesting and relevant intervention. A few suggestions and comments have been provided.

Line 45: edit “immediately after childbirth”

Line 85 – 87: general editing for clarity. Physical activity? Remove ‘and’ after ‘activity’.

Line 165-167: Indicates the app allows self-monitoring and real-time feedback, but line 352 – 354 mentions that the current app does not include ‘self-review options’. Could you please clarify what/when the app would allow participants to see their data? I may have overlooked it, but how often would the 12 sessions take place?

In follow up to above, line 354 – 357: Agreed that data visualization would be helpful addition!

General comment: Could this tool be linked with wearable activity monitors that individuals may already have (or be supplied as part of an intervention) to help monitor sleep and activity more accurately than self-report? Light monitoring as well?

Reviewer 3 Report

This is an interesting study on ClockWork intervention and associated digital tools for improving cardiometabolic health-related behaviors. The authors have an ambitious approach, stretching the use of digital monitoring tools in addressing complications with obese postpartum individuals. This, however, results in a structured text with an appreciable but occasionally overly wordy, which can be justified while presenting a topical area that is gaining momentum in the AI field. The authors have taken a meticulous effort to present their findings and provide all necessary information in the form of figures and data etc. But the main concern is with the very low sample size, which limits the scope and importance of this research. My suggestion to the authors is to add this to the limitations. Other minor issues need to be addressed: The authors have not mentioned anything about the exclusion criteria in this study, and also it would be better if the authors presented the demographic data as a table; Can the authors specify how long this study was conducted; Kindly make sure that all wordings are visible in table 2.
